# Endemic High-Risk Clone ST277 Is Related to the Spread of SPM-1-Producing *Pseudomonas aeruginosa* during the COVID-19 Pandemic Period in Northern Brazil

**DOI:** 10.3390/microorganisms11082069

**Published:** 2023-08-11

**Authors:** Pabllo Antonny Silva Dos Santos, Yan Corrêa Rodrigues, Davi Josué Marcon, Amália Raiana Fonseca Lobato, Thalyta Braga Cazuza, Maria Isabel Montoril Gouveia, Marcos Jessé Abrahão Silva, Alex Brito Souza, Luana Nepomuceno Gondim Costa Lima, Ana Judith Pires Garcia Quaresma, Danielle Murici Brasiliense, Karla Valéria Batista Lima

**Affiliations:** 1Program in Parasitic Biology in the Amazon Region (PPGBPA), State University of Pará (UEPA), Tv. Perebebuí, 2623-Marco, Belém 66087-662, PA, Brazil; antonnypabllo@gmail.com (P.A.S.D.S.); davijosuemarcon@gmail.com (D.J.M.); luanalima@iec.gov.br (L.N.G.C.L.); daniellemurici@iec.gov.br (D.M.B.); 2Bacteriology and Mycology Section, Evandro Chagas Institute (SABMI/IEC), Ministry of Health, Ananindeua 67030-000, PA, Brazil; amalialobato@iec.gov.br (A.R.F.L.); thatacazuza@gmail.com (T.B.C.); isabelmontoril13@gmail.com (M.I.M.G.); jesseabrahao10@gmail.com (M.J.A.S.); alexbritosouza@yahoo.com.br (A.B.S.); anaquaresma@iec.gov.br (A.J.P.G.Q.); 3Program in Epidemiology and Health Surveillance (PPGEVS), Evandro Chagas Institute (IEC), Ministry of Health, Ananindeua 67030-000, PA, Brazil; 4Department of Natural Science, State University of Pará (DCNA/UEPA), Belém 66050-540, PA, Brazil

**Keywords:** *Pseudomonas aeruginosa*, SPM-1, northern region, virulence, resistance

## Abstract

*Pseudomonas aeruginosa* is a high-priority bacterial agent that causes healthcare-acquired infections (HAIs), which often leads to serious infections and poor prognosis in vulnerable patients. Its increasing resistance to antimicrobials, associated with SPM production, is a case of public health concern. Therefore, this study aims to determine the antimicrobial resistance, virulence, and genotyping features of *P. aeruginosa* strains producing SPM-1 in the Northern region of Brazil. To determine the presence of virulence and resistance genes, the PCR technique was used. For the susceptibility profile of antimicrobials, the Kirby–Bauer disk diffusion method was performed on Mueller–Hinton agar. The MLST technique was used to define the ST of the isolates. The *exoS^+^/exoU^−^* virulotype was standard for all strains, with the *aprA*, *lasA*, *toxA*, *exoS*, *exoT,* and *exoY* genes as the most prevalent. All the isolates showed an MDR or XDR profile against the six classes of antimicrobials tested. HRC ST277 played a major role in spreading the SPM-1-producing *P. aeruginosa* strains.

## 1. Introduction

*Pseudomonas aeruginosa* is a high-priority bacterial agent that causes healthcare-acquired infections (HAIs), which often lead to serious infections and poor prognosis in vulnerable patients, such as those who are in intensive care units (ICUs); those who have weakened immune systems, have undergone surgery, or have a history of inappropriate antibiotic use or severe burns; and those who have cystic fibrosis (CF), causing chronic lung colonization [1,2,3,4]. Globally, multi-drug and extensively resistant (MDR/XDR) strains of *P. aeruginosa* posing a difficult-to-treat resistance (DTR) phenotype have emerged in different clinical, hospital, and even environmental settings. These strains are of particular concern due to difficulties and limitations in treatment and their association with a high virulence potential, which can lead to severe and prolonged infections, and increased treatment costs, length of hospital stay, and patient mortality [5,6,7].

As a versatile opportunistic pathogen, *P. aeruginosa* is capable of causing both acute and chronic infections. Its pathogenic profile stems from the large and variable arsenal of virulence factors and antibiotic resistance determinants contained in the *P. aeruginosa* genome of several strains, with remarkable metabolic flexibility and the ability to adapt to multiple conditions, including the host immune response [8,9,10]. Virulence products passively produced and secreted by bacterial cells are generally associated with adhesion, nutrient acquisition, and regulation, including pigments with siderophore activity and O-polysaccharide (OPS), whereas products actively secreted by secretion systems, such as the type I secretion system (*T1SS*), type II secretion system (*T2SS*), and type III secretion system (*T3SS*), are associated with tissue invasion and evasion of host defenses [11,12,13]. Among these, the virulotyping of *P. aeruginosa* strains by detecting *exoS*/*exoU* genes and OPS serotyping has been widely applied and recommended due to its association with the clinical progress of patients, antimicrobial resistance (AMR), and vaccine targets [14,15,16,17].

Concerning AMR, the production of carbapenemases has been pointed out as one of the main causes of carbapenem resistance among several bacterial pathogen species. Indeed, carbapenem-resistant *P. aeruginosa* (CR-PA) has been described as a priority pathogen by the World Health Organization (WHO) and several other health agencies [1,18,19,20]. Among the carbapenemases, metallo-β-lactamases (MβLs) are of particular interest and concern due to several factors, such as their ability to hydrolyze and provide resistance to virtually all β-lactam antibiotics, the limitations and unavailability of clinically useful MβLs inhibitors, the rapid rate at which new variants are isolated, the transferability of their coding genes, and their ubiquity, as there are reports of isolates from both hospitals and environmental sources [21,22]. The São Paulo metallo-β-lactamase (SPM-1) is an important determinant of carbapenem resistance and non-susceptibility phenotypes in *P. aeruginosa* isolates in Brazil. In different Brazilian geographic regions, the dissemination of SPM-1-producing *P. aeruginosa* is associated with the endemic clone, ST277, which may be related to the high and increasing rates of carbapenem resistance reported [23,24,25,26].

Several recent reports have described an increase in MDR/XDR organisms during the COVID-19 pandemic [27,28]. In the current pandemic healthcare emergency, sentinel reports have shown that secondary infections were present in up to 30% of critically ill patients, and these infections were shown to markedly decrease the survival of patients with COVID-19. MDR/XDR-CR-PA was one of the most commonly reported antibiotic-resistant bacterial species in COVID-19 patients admitted to the ICUs [29,30]. The production of SPM-1 has been proven as a key antimicrobial mechanism in Brazilian *P. aeruginosa* strains, and with the pandemic situation of COVID-19 and rampant use of antibiotics for the treatment of secondary infections, a relevant increase in CR-PA isolates harboring the *bla_SPM-1_* gene has been observed in health institutions in northern Brazil. Additionally, previous reports in the state of Pará by our research group have revealed the presence of *P. aeruginosa* harboring the *bla_SPM-1_* gene, which causes complicated infections and is a genotypic marker of high virulence [31,32]. Thus, this study aims to explore the AMR, virulence, and genotypic features of SPM-1-producing *P. aeruginosa* recovered from the pre-pandemic period in healthcare institutions in the states of Pará (PA) and Acre (AC), in the northern Brazilian Amazon region.

## 2. Materials and Methods

### 2.1. Bacterial Isolates

This is a cross-sectional and descriptive study aiming to provide data on the SPM-1-producing-*P. aeruginosa* isolates received at a reference center—the Special Pathogens Laboratory, Bacteriology and Mycology Evandro Chagas Institute (LabPate/SABMI/IEC)—for the routine surveillance of antimicrobial resistance. Since mid-2017, LabPate/SABMI/IEC has been acting in the antimicrobial resistance surveillance flow routine by confirming and detecting AMR mechanisms in bacterial isolates from public and private hospitals in the states of Pará (PA) and Acre (AC), northern Brazil. For the present study, 34 non-repeatable isolates of *P. aeruginosa* were obtained from various biological sample sources of patients admitted to healthcare services from 2018 to 2021, with suspected infection and/or colonization by MDR/XDR microorganisms and production of carbapenemases (resistance to carbapenems). All the isolates were identified using the Vitek-2 automated system at a routine hospital (BioMérieux). Subsequently, the isolates were sent to Evandro Chagas Institute for further analysis.

### 2.2. Phenotypic and Molecular Assays Associated with Antimicrobial Susceptibility and Genetic Variant Definition of bla_SPM-1_

Antimicrobial susceptibility testing (ATS) was performed by applying the Kirby–Bauer disk diffusion method on Mueller–Hinton Agar (MHA) for 12 antimicrobials belonging to six (06) different classes: piperacillin, piperacillin + tazobactam, and ticarcillin/clavulanic acid (penicillin + β-lactamase inhibitor class); ceftazidime and cefepime (cephalosporins class); aztreonam (monobactams class); imipenem (carbapenems class); gentamicin, tobramycin, and amikacin (aminoglycosides class); and ciprofloxacin and ofloxacin (fluoroquinolone class). The results were interpreted according to the criteria and breakpoints of Clinical and Laboratory Standards Institute, where isolates were classified as susceptible (S), intermediate (I), and resistant (R) [33,34]. Additionally, *P. aeruginosa* isolates were phenotypically classified based on their propensity to be MDR when they were resistant to ≥1 drug in ≥3 antimicrobial classes; XDR when they were not susceptible to 1 agent in all antimicrobial classes tested, except ≤2, according to the criteria described by Magiorakos et al. [35] and Mulet et al. [36]; and DTR based on the susceptibility results with ceftazidime, cefepime, imipenem, ciprofloxacin, and ofloxacin, as described by Kadri et al. [7].

Bacterial genomic DNA was obtained from a single overnight grown colony of *P. aeruginosa* cultures via the boil-and-freeze method and using the commercial PureLink™ Genomic DNA Mini Kit (Thermo Fisher Scientific, São Paulo, Brazil), following the manufacturer’s recommendations. The genomic DNA obtained was quantified using a Picodrop PICO100 spectrophotometer (Picodrop Limited, Hinxton, UK) and concentrations set between 25–50 ng/μL were used for all molecular assays. The detection of AMR genes encoding carbapenemase *bla_SPM_*, *bla_IMP_*, *bla_VIM_*, *bla_NDM_*, *bla_KPC_*, and *bla_OXA-48_* was performed via PCR in a Veriti thermal cycler (Applied Biosystem, Foster City, CA, USA) as described [37]. Visualization of PCR products was performed via electrophoresis in a 1.5% agarose gel at 110 V for 45 min in TAE 1× buffer (89 nM Tris-borate and 2 mM EDTA pH 8.0). DNA ladder 1 Kb (Invitrogen™, Carlsbad, CA, USA)) was used as molecular weight marker, gel stained with SyberSafe (Invitrogen™, Carlsbad, CA, USA)), and differentiation of bands visualized under ultraviolet light.

For determination of the *bla_SPM_* variant, the PCR products were direct sequenced bidirectionally using the Big Dye Terminator v3.1 kit on the ABI Prism 3100 or 3500XL Genetic Analyzer platform (Applied Biosystems, Foster City, CA, USA), and the sequences obtained were compared with those available in the BLAST database (https://blast.ncbi.nlm.nih.gov/Blast.cgi (accessed on 6 June 2023)).

### 2.3. Molecular and Phenotypic Detection of Virulence-Related Factors

The detection of invasion-related genes belonging to the *T1SS*, *T2SS* and *T3SS* was performed via PCR in a Veriti thermal cycler (Applied Biosystem, Foster City, CA, USA) according to the protocol described by Rodrigues et al. [32]. Visualization of PCR products was performed via 1.5% agarose gel electrophoresis at 110 V for 45 min in TAE 1× buffer (89 nM Tris-borate and 2 mM EDTA pH 8.0). As molecular weight marker, 1 Kb DNA ladder (Invitrogen™) was used, gel stained with SyberSafe (Invitrogen™, Carlsbad, CA, USA)) and differentiation of bands visualized under ultraviolet light. In addition, the pigment production and mucoid phenotype of *P. aeruginosa* isolates were verified by observing bacterial growth on MHA agar plates and slants.

### 2.4. Molecular Typing by Multilocus Sequencing Typing–MLST

The MLST genotyping procedure followed the protocol outlined by Curran et al. [38], with modifications by using new design primers, except for *aroE* gene (Appendix A). In brief, the Veriti thermocycler (Applied Biosystems, Foster City, CA, USA) was used to amplify via PCR the seven housekeeping genes constituting the scheme (*acsA*, *aroE*, *guaA*, *mutL*, *nuoD*, *ppsA*, and *trpE*). The resulting reaction products were sequenced bidirectionally using Big Dye Terminator v3.1 chemistry on the ABI Prism 3100 or 3500XL Genetic Analyzer platforms (Applied Biosystems, Foster City, CA, USA). The obtained results were compared and matched to the data available at the PubMLST database (http://pubmlst.org/paeruginosa (accessed on 6 June 2023)) to determine the allelic profiles and sequence types (STs).

### 2.5. Whole-Genome Sequencing (WGS) and Bioinformatics Analysis

Libraries were prepared from the previously extracted DNA using the Nextera XT kit (Illumina, San Diego, CA, USA) with the addition of I5 and i7 indexes, according to the manufacturers’ protocol. The quality of the libraries was verified using the Bioanalyzer High Sensitivity DNA Analysis kit (Agilent™, Santa Clara, CA, USA) and quantified using the High Sensitivity Double Strand DNA Quibit kit (Invitrogen™, Carlsbad, CA, USA)). Subsequently, the libraries were added into a pool and sequenced with the 2 × 151 paired-end protocol on Illumina nextseq 550 using Mid Output (Illumina™) reagent cartridges and flow cells at the Arbovirology section of the Instituto Evandro Chagas.

The quality of the reads was checked using the fastqc v0.11.9 tool, treated using the fastp v0.23.2 tool to remove low quality reads and remove adapters. Subsequently, genome assembly was performed using the spades tool v3.15.3 based on the reference strains for *P. aeruginosa* CCBH4851 (NZ_CP021380.2), which belongs to clone ST277 reported as cause of endemic outbreak in Brazil in 2008 [39]. After assembly, the scaffolds were evaluated using the quast software (v 5.2.0) and submitted to the bactopia v2.2 pipeline for annotation using the prokka tool v1.14.6, resistance prediction using amrfinder v3.10.45. The modular tools of the bactopia pipeline were also used for downstream analysis: abricate for searching resistance genes, amrfinderplus for predicting resistance and proteins, MLST typing was predicted by searching for sequence in the PubMLST database, pasty for predicting *P. aeruginosa* serogroup, and plasmidfinder for predicting plasmid presence in sequencing. Finally, the annotated genomes produced by Bactopia were finally submitted type Strain Genome Server (TGYS) [40] web server for whole-genome similarity, clusterization and phylogenetic inference.

### 2.6. Ethical Considerations

The present study is in accordance with the principles of the Declaration of Helsinki and the terms of the CNS Resolution No. 466/2012 of the National Health Council. Since this is an experimental study, which used stored and provided samples by the institutions involved, without any contact and possibility of identifying the respective patients, the project did not need to be referred to the Ethics Committee on Research Involving Human Beings.

## 3. Results

### 3.1. Antimicrobial-Susceptibility-Related Features

The AST results revealed that all included *P. aeruginosa* isolates were resistant to carbapenems (IMP, 34/34–100.0%), followed by significant non-susceptibility to antipseudomonal fluoroquinolones (OFX, 33/34–97.1%; CIP, 31/34–91.7%), antipseudomonal penicillin + β -lactamase inhibitors (PRL and TTC, 32/34–94.1%), antipseudomonal cephalosporins (CAZ and FEP 31/34–91.7%), and aminoglycosides (GEN, 32/34–94.1%; TOB, 30/34 and AMK 30/34–88.2%). Antagonistically, ATM and TZP were considered to be the most effective antimicrobials with 61.7% (21/34) and 26.5% (9/34) of sensitive isolates, respectively. According the susceptibility classification, 64.7% (22/34) were phenotypically classified as MDR, 35.3% (12/34) as XDR, and 26.5% (9/34) as DTR (Table 1).

Molecular detection of carbapenemase genes and sequencing confirmed that all included CR-PA harbored the *bla_SPM-1_* variant (34/34–100%). The *bla_IMP_*, *bla_VIM_*, *bla_NDM_*, *bla_KPC_*, and *bla_OXA-48_* genes were not detected (Table 1 and Table 2).

### 3.2. Virulence-Related Features

Most of the *T1SS*, *T2SS*, and *T3SS* virulence genes (*aprA*, *lasA*, *toxA*, *exoS*, *exoT*, and *exoY*) were homogenously detected among evaluated isolates (34/34–100.0%), while a sample only (1/34–3.0%) was negative for the *lasB* gene. Additionally, all isolates (34/34–100.0%) were related to the invasive virulotype (*exoS*^+^/*exoU*^−^), as the *exoU* gene was absent. As for the mucoid characteristic and pigment production by the colonies, 44.1% (15/34) of the samples were presenting a mucoid-like feature, and positivity for both pyocyanine and pyoverdine pigments (Table 2).

### 3.3. Genotyping by MLST Data

From the pool of *P. aeruginosa* isolates presenting MDR phenotypes, nine (9) randomly selected isolates were subjected to molecular typing via MLST, revealing that all nine MDR-*P. aeruginosa* isolates belonged to the high-risk clone (HRC) and endemic clone ST277 determined by the combination of the seven housekeeping genes used in the MLST scheme for *P. aeruginosa* (*acsA* 39, *aroE* 5, *guaA* 9, *mutL* 11, *nuoD* 27, *ppsA* 5, and *trpE* 2) (Table 2).

### 3.4. WGS Data Results

From the pool of *P. aeruginosa* isolates presenting XDR phenotypes, 10 randomly selected isolates were subjected to WGS analysis. Most of the XDR samples (9/10) presented similarity (dDDH-d_0_) between 99.9–100% and had MLST associated with the HRC ST277. The following set of antimicrobial resistance genes was detected: *aac(6′)-Ib1*, *aadA7*, *aph(3′)-IIb*, *bla_OXA-494_*, *bla_OXA-56_*, *bla_PDC-374_*, *bla_SPM-_1*, *catB7*, *cmx*, *crpP*, *fosA-354827590*, *rmtD1*, and *sul*, which indicated resistance to the multiple antimicrobial classes, such as: carbapenems, cephalosporins, chloramphenicol, fluoroquinolones, fosfomycin, gentamicin, kanamycin, streptomycin and sulfonamide. Interestingly, the sample, 57,508, presented similarity between 99.1–99.5% when compared to the other nine samples, and 99.3% similarity when compared to CCBH485 strain. This sample was also found to belong to ST2711 (MLST confirmed via sanger sequencing). A similar set of resistance genes was found, however, with presence of the *bla_OXA-50_* gene instead of the *bla-_OXA494_* (Figure 1). Via serotyping prediction, all samples were related to the O2 serogroup.

## 4. Discussion

Recently, the rapid emergence of CR-PA strains has become prominent in scientific interest and epidemiological surveillance, mainly due to the dissemination of MβLs that break down antibiotic compounds that are commonly used as a last-resort treatment to serious infections, rendering penicillin, cephalosporins, and carbapenems ineffective. This scenario is the result of several factors, including the overuse and misuse of antibiotics, and the poor infection control practices in healthcare settings. Certainly, the COVID-19 pandemic has also placed a tremendous pressure on healthcare systems worldwide, as critically ill patients were at increased risk for secondary bacterial infections associated with MDR/XDR/DTR strains, including MβL-producing-*P. aeruginosa*. In the present investigation, we report the spread of SPM-1-producing-*P. aeruginosa* strains mostly associated with the HRC ST277, and detected since the pre and early COVID-19 pandemic period in healthcare institutions in northern Brazilian.

Worrying rates of AMR associated with XDR/MDR/DTR *P. aeruginosa* isolates have been reported in the last decade, as demonstrated by Jean et al. [41] in Taiwan, where the AMR rate in 2015 was less than 18.0%, while in the following years (2016 and 2018), the rate increased to 19.7% and 27.5%, respectively. A study conducted in Spain reported that 17.0% of *P. aeruginosa* infections were caused by XDR strains, and high rates of over 30.0% of CR-PA were linked to hospital-acquired pneumonia (HAP) as reported in many European Union states since 2015 [42,43]. Additionally, DTR among *P. aeruginosa* were related to almost 8.0% of isolates causing BSIs [44]. Despite this, there is still scarce global information on the prevalence of MDR/XDR/DTR-*P. aeruginosa* [20]. Further, due to the similarity of symptoms between hospitalized patients with SARS-CoV-2 infection and those with hospital-acquired and ventilator-associated pneumonia, it is a common practice to administer broad-spectrum antibiotics as empirical treatments [45]. According to a review conducted by Fattorini et al. [29], 476 out of 539 patients (88.3%) diagnosed with COVID-19 received broad-spectrum antibiotics, such as expanded-spectrum cephalosporins (e.g., ceftriaxone, ceftazidime, and cefepime), fluoroquinolones, and carbapenems. Consequently, the use of antibiotics has significantly increased in many healthcare settings globally during this period [46].

As per national data by the Brazilian National Health Surveillance Agency (ANVISA), from 2018 to 2021 in adult ICUs, CR-PA was the third-most-detected bacterial pathogen related to BSIs and urinary-tract infections (UTIs), and demonstrated carbapenem resistance rates from 30.9% to 41.4%, and from 41.7% to 43.0%, respectively [47,48,49,50,51]. Worryingly, it is relevant to emphasize the staggering increase in the number of *P. aeruginosa* isolates causing BSIs in 2021 (pandemic-period), totaling 3,845 cases, a remarkable 168.1% surge compared to 2019 (pre-pandemic period), which recorded only 1432 cases. Surely, the resistance phenotypes of the CR-PA in this study, which included MDR/XDR/DTR isolates, reflect this worrisome scenario, further complicated by the presence of SPM-producing isolates. Finally, such findings also align with our research group’s previous data, in which Rodrigues et al. [32] documented the early spread of MDR/XDR CR-PA within local ICUs in the state of PA from 2010 to 2013.

The monobactam antibiotic ATM has presented potential in the treatment of infections caused by MDR/XDR CR-PA [52,53]. In the present report, ATM has been indicated as an effective antimicrobial against CR-PA, with a resistance rate of only 37.1%. This sensitivity profile can be attributed to the fact that the antibiotic is not broken down by SPM. Studies conducted worldwide and in Brazil have reported similar findings, suggesting the strong efficacy of ATM against CR-PA [54,55]. However, it is noteworthy that resistance to ATM was observed in some isolates, pointing out the presence of other AMR mechanisms, such as mutations observed in *mexAB-oprM* efflux system [56]. Further investigations are needed to fully understand the role of ATM and its potential strategies in the management of CR-PA infections [57].

Results obtained through the WGS analysis of the 10 XDR SPM-1-producing *P. aeruginosa* allowed further insights into the AMR mechanisms presented in such strains, in which the *aac(6′)-Ib’*, *aadA7*, *aph(3′)-IIb*, *bla_OXA-56_*, *bla_PDC-374_*, *bla_SPM-1_*, *catB7*, *cmx*, *crpP*, *fosA-354827590* and *rmtD1* markers were commonly found. For the *bla_OXA-494_* gene, only one sample was negative; in contrast, for the *bla_OXA-50_* gene, only one sample was positive. This bacterial resistome echoes the findings in the Brazilian study published by Galetti et al. (2018), where in genomic analysis of 13 different *P. aeruginosa* strains belonging to ST277 revealed a highly conserved resistome (*bla_SPM-1_*, *rmtD*, *aacA4*, *aadA7*, *bla_OXA-56_*, *bla_OXA-396_*, *bla_PAO_*, *aph(3′)-IIb*, *aac(6′)Ib-cr*, *crpP*, *catB7*, *cmx*, and *fosA*), playing an important role in the persistence of this clone in infections occurring in Brazilian hospitals. The *bla_OXA_* gene variants are considered as naturally occurring in the *P. aeruginosa* genome, and its high prevalence indicates a potential horizontal transfer in which class D β-lactamases can be introduced by other co-habituating bacterial species [58,59]. According to Horcajada et al. [1] and Nicolau [60], the *bla_OXA-50_* gene plays an important role in *P. aeruginosa* resistance, since classical β-lactamase inhibitors show weak activity against it. Indeed, kinetic analysis of β-Lactams hydrolysis by *OXA-50* variants of *P. aeruginosa* demonstrated that chromosomally encoded AMR mechanisms mainly provided weak carbapenemase activity, but may act synergically [61]. Among the aminoglycoside-modifying enzymes presented, the *aac(6′)* acetyltransferase is one of the most frequently described, conferring resistance to both tobramycin and amikacin, or tobramycin alone [1,62,63].

To fuel its pathogenicity, *P. aeruginosa* possesses an array of virulence factors that enable the colonization, invasion, and persistence within host tissues, often leading to acute and chronic challenging-to-treat infections. Gaining a comprehensive understanding of these virulence mechanisms is imperative for the development of effective strategies to manage *P. aeruginosa* infections [13]. In relation to presence of pigments like pyocyanin and pyoverdine, it has been implicated in exacerbating infections as these pigments sequester iron from host cells, serving the metabolic needs of the bacterium, and consequently intensifying the infection and pathogenesis [64]. A study conducted by Fothergill et al. [65] reported pyocyanin production in *P. aeruginosa* isolates ranging from 41.3% to 81.5%, findings consistent with the data obtained in the current investigation, where pyocyanin production among isolates was of 42.9%. With regard to pyoverdin, Prado et al. [66] observed pyoverdin production in over 74.0% of clinical strains, while Silva et al. [67] found that more than 90.0% of the isolates investigated exhibited pyoverdin production. Interestingly, the present study recorded a pyoverdine production rate of 42.9%, which contrasts with the previous findings. In this study, *aprA*, a gene belonging to *T1SS*, and *lasA* and *lasB* genes belonging to *T2SS*, showed high positive occurrence. Other studies with SPM-1-producing *P. aeruginosa* also reported a strong presence of these virulence genes, as in the studies by Adonizio et al. [68] and Silva et al. [67].

In addition, the translocation of up to four cytotoxic effector proteins by the *T3SS* is responsible for distinct tissue injury to the host, with *exoU* having a higher impact on bacterial virulence [11]. The distribution of the genes encoding these cytotoxins is not uniform among *P. aeruginosa* strains, and some of them, particularly *exoS* and *exoU*, are almost mutually exclusive [69]. In fact, a large, multicenter study conducted in Spain revealed that the *exoU*^+^*/exoS*^−^ genotype was an independent risk factor for early mortality in *P. aeruginosa* BSIs, and was negatively linked to XDR profiles [14].Thus, the *T3SS* factor is an important differential factor that needs to be considered when analyzing virulence and clinical outcomes associated with HRC [70]. Results on the present study highlight the fact that all evaluated strains were related to the invasive virulotype *(exoS*^+^*/exoU*^−^*)*, genotypic virulence profile usually observed among MDR/XDR *P. aeruginosa* strains, as *exoU* carriage along with several AMR mechanisms may pose a fitness cost to bacterial cell [71,72,73]. Furthermore, all 10 samples analyzed belonged to serogroup O2. According to Stanislavsky [74], polysaccharide O (OPS), the most variable region of the lipopolysaccharide (LPS), is of major relevance in the virulence and is responsible for conferring serogroup specificity. According to Donta et al. [75], serogroup O2, along with serogroups O1, O3, O4, O5, O6, O7, O10, and O16, accounts for 90% of bacteriemic strains of *P. aeruginosa*. In a study by Nasrin et al. [17], serotype O2, along with serotypes O5, O16, O18, and O20, were among the most common, which corroborates with the findings of the present study.

Global epidemiology data further highlight a small geographic spread of *bla_SPM-1_* strains when compared to its endemicity in Brazil, with rare reports of this variant in countries such as Iran [76,77], UK [78], Chile [79], Egypt [80], and the USA [81,82,83]. In Brazil, the clonal expansion of SPM-1-producing *P. aeruginosa* strains is related to the HRC ST277, with its detection in all Brazilian regions, including São Paulo [25], Rio de Janeiro [84], Paraná [85], Porto Alegre [86], Minas Gerais [87], and Pará [31,32], showing its dissemination potential, high adaptability, and establishment as an international clone. In the present report, the MLST genotyping revealed that 18 strains with MDR/XDR/DTR characteristics belonged to the ST277 lineage, and one related to the ST2711, which, to the best of our knowledge, is the first report of *bla_SPM-1_* in another clone than the ST277. This finding also supports the limited genetic diversity of SPM-1-producing *P. aeruginosa*, and indicates the possible occurrence of an outbreak, with recent clonal expansion probably related to the high selective pressure on healthcare institutions in northern Brazil. The clonal expansion of such strain also raises concerns regarding the potential dissemination of AMR gene, and the limited effectiveness of conventional treatment options. Further, when comparing the genomic phylogenetic inference, we highlight the distance between sample 57508/ST2711 and the other samples, being the most distant sample when compared to CCBH4851; the remaining samples were clustered as a possible transmission chain due elevated similarity (above 99.9%). Further investigation is needed to understand the underlying mechanisms driving the persistence and spread of these particular STs in the clinical or environmental settings.

As the pandemic spread, hospitals globally observed an increase in patients infected with COVID-19, a situation requiring major adjustments in healthcare systems and infrastructure, especially in infection control and antimicrobial management programs [88]. In this regard, some reports indicate that the indiscriminate use of antibiotics determined by the therapeutic challenges in combating the pandemic has resulted in increased AMR rates, especially related to individuals infected with *P. aeruginosa* hospitalized in ICUs [89]. Unfortunately, less robust healthcare systems, such as those in the Latin American and Asian countries, where AMR rates are dangerously high and antimicrobial stewardship programs are just beginning to be implemented, are adjusting their response to the pandemic to varying degrees [90,91,92,93]. Regrettably, these circumstances create the so-called “perfect storm” for an accelerated evolution of AMR, especially in clinically important strains, such as *P. aeruginosa* [94]. The present study is one of first in Brazil to thoroughly report data on AMR after the COVID-19, pandemic reflecting a comparative perspective between studies conducted before the pandemic, where the peak detection of SPM-1-producing *P. aeruginosa* strains occurred between 2008 and 2015 [32,95,96,97], and in the post-pandemic context, as with results in the current study, a re-emergence and the possibility of an outbreak of SPM-1-producing *P. aeruginosa* was observed.

The present study is not without its limitations. Firstly, a notable limitation was the loss of isolates during the culture process, which may have resulted in an incomplete dataset. Additionally, our laboratory faced the constraint of unavailability of certain essential testing disks for evaluating classical antipseudomonal drugs such as meropenem and colistin, and novel antibiotics including cefiderocol, ceftazidime-avibactam, and ceftolozane-tazobactam. Another limitation stems from the lack of comprehensive data regarding the origin and specific wards from which the *P. aeruginosa* isolates were recovered, limiting our ability to assess potential associations between strain characteristics and clinical settings, and outbreak investigation. Lastly, all included samples could not be genotyped via MLST and WGS due to a lack of necessary reagents, which could have provided valuable insights into genetic relatedness and transmission patterns. These limitations should be taken into account when interpreting the findings and highlight areas for further investigation and improvement in future studies.

## 5. Conclusions

The CR-PA isolates included in this study showed a high prevalence of virulence genes, where among them, *aprA*, *lasA*, *toxA*, *exoS*, *exoT*, and *exoY* were positive in all strains, suggesting a high pathogenicity capacity. The *exoS*^+^*/exoU*^−^ virulotype was standard in all isolates, indicating an invasive characteristic. As for the phenotypic profile of resistance, all strains showed either MDR or XDR, in addition to a pool of DTR isolates, thus posing a challenge regarding the management and treatment of patients infected with *P. aeruginosa* producing SPM. Additionally, results obtained through MLST and WGS revealed the major role of the HRC ST277 in spreading SPM-1-producing strains, in addition to the novel report of the *bla_SPM-1_* variant in the clone, ST2711, and a conserved resistome.

## Figures and Tables

**Figure 1 microorganisms-11-02069-f001:**
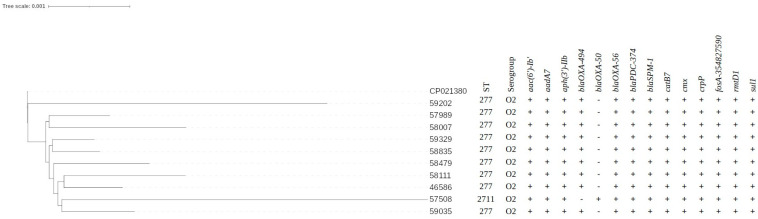
Similarity and genomic characteristics of XDR SPM-1-producing *P. aeruginosa* isolated during pandemic period in northern Brazil. (+: positive, −: negative).

**Table 1 microorganisms-11-02069-t001:** Antimicrobial susceptibility phenotypes of SPM-1-producing *P. aeruginosa* isolates in northern Brazil.

ID	PRL	TZP	TTC	CAZ	FEP	ATM	IMP	GEN	TOB	AMK	CIP	OFX	Susceptibility Phenotype
46586	I	I	I	R	R	R	R	R	R	R	R	R	XDR #
46716	I	S	R	R	R	S	R	I	S	S	R	R	MDR
54178	S	S	I	R	R	S	R	R	R	R	R	R	MDR
56158	I	R	I	R	R	S	R	R	R	R	R	R	MDR
56572	R	S	S	S	S	R	R	S	R	S	S	R	MDR
57415	I	S	S	S	S	S	R	S	S	R	R	R	MDR
57508	R	S	I	I	S	I	R	R	S	S	S	R	XDR
57564	I	I	R	R	R	S	R	R	R	R	R	R	MDR
57568	R	R	R	R	R	I	R	R	R	R	R	R	XDR #
57654	R	I	R	R	R	S	R	R	R	R	R	R	MDR
57716	I	R	R	R	R	S	R	R	R	R	R	R	MDR
57729	R	I	R	R	R	S	R	R	R	R	R	R	MDR
57863	R	I	R	R	R	S	R	R	R	R	R	R	MDR
57877	I	S	R	R	R	I	R	R	R	R	R	R	XDR #
57884	R	I	R	R	R	S	R	R	R	R	R	R	MDR
57989	R	I	R	R	R	I	R	R	R	R	R	R	XDR #
58005	R	I	R	R	R	S	R	R	R	R	R	R	MDR
58007	I	I	R	R	R	I	R	R	R	R	R	R	XDR #
58111	R	I	R	R	R	I	R	R	R	R	R	R	XDR #
58218	I	I	R	R	R	S	R	R	R	R	R	R	MDR
58276	R	I	R	R	R	S	R	R	R	R	R	R	MDR
58479	R	S	R	R	R	R	R	R	R	R	R	R	XDR #
58482	I	I	R	R	R	S	R	R	S	S	R	R	MDR
58608	R	R	R	R	R	S	R	R	R	R	R	R	MDR
58739	S	R	R	S	R	S	R	R	S	R	S	S	MDR
58798	R	R	R	R	R	S	R	R	R	R	R	R	MDR
58820	I	I	R	R	R	S	R	R	R	R	R	R	MDR
58835	I	S	R	R	R	R	R	R	R	R	R	R	XDR #
58924	R	I	R	R	R	S	R	R	R	R	R	R	MDR
59035	R	I	R	R	R	R	R	R	R	R	R	R	XDR #
59183	R	I	R	R	R	S	R	R	R	R	R	R	MDR
59202	R	I	R	R	R	R	R	R	R	R	R	R	XDR #
59233	I	S	R	R	R	S	R	R	R	R	R	R	MDR
59329	R	I	R	R	R	R	R	R	R	R	R	R	XDR #

PRL (piperacillin); TZP (piperacillin + tazobactam); TTC (ticarcillin-clavulanic Acid); CAZ (ceftazidime); FEP (cefepime); ATM (aztreonam); IMP (imipenem); GEN (gentamicin); TOB (tobramycin) AMK (amikacin); CIP (ciprofloxacin); OFX (ofloxacin); # DTR isolates.

**Table 2 microorganisms-11-02069-t002:** AMR, virulence and genotypic features of SPM-1-producing *P. aeruginosa* isolates in northern Brazil.

ID	Date at LabPate/IEC	Biological Source	Origin	Resistance Phenotype	ST	*bla_SPM−1_*	*bla_IMP_*	*bla_VIM_*	*bla_NDM_*	*bla_KPC_*	*bla_OXA−48_*	*aprA*	*lasA*	*lasB*	*toxA*	*exoS*	*exoU*	*exoT*	*exoY*	Mucoid	Pyoverdine	Pyocyanine
46586	31 July 2018	Urine	PI/AC	XDR	277 *	+	−	−	−	−	−	+	+	+	+	+	−	+	+	+	+	+
46716	8 August 2018	Urine	PI/PA	MDR	277	+	−	−	−	−	−	+	+	+	+	+	−	+	+	−	−	−
54178	6 November 2019	TS	PI/AC	MDR	277	+	−	−	−	−	−	+	+	+	+	+	−	+	+	−	−	−
56158	27 March 2020	TS	PI/AC	MDR	277	+	−	−	−	−	−	+	+	+	+	+	−	+	+	+	+	+
56572	21 May 2020	TS	PI/AC	MDR	277	+	−	−	−	−	−	+	+	+	+	+	−	+	+	+	+	+
57415	10 February 2021	Urine	PR/PA	MDR	277	+	−	−	−	−	−	+	+	+	+	+	−	+	+	−	−	−
57508	3 March 2021	TS	PI/PA	XDR	2711 *^§^	+	−	−	−	−	−	+	+	+	+	+	−	+	+	+	+	+
57564	22 March 2021	Urine	PR/PA	MDR	277	+	−	−	−	−	−	+	+	+	+	+	−	+	+	+	+	+
57568	22 March 2021	TS	PR/PA	XDR		+	−	−	−	−	−	+	+	+	+	+	−	+	+	+	+	+
57654	6 April 2021	TS	PR/PA	MDR	277	+	−	−	−	−	−	+	+	+	+	+	−	+	+	−	−	−
57716	6 May 2021	TS	NI	MDR	277	+	−	−	−	−	−	+	+	+	+	+	−	+	+	−	−	−
57729	6 May 2021	TS	NI	MDR	277	+	−	−	−	−	−	+	+	+	+	+	−	+	+	+	+	+
57863	24 May 2021	Urine	NI	MDR		+	−	−	−	−	−	+	+	+	+	+	−	+	+	−	−	−
57877	24 May 2021	Blood	NI	XDR		+	−	−	−	−	−	+	+	+	+	+	−	+	+	−	−	−
57884	24 May 2021	TS	NI	MDR		+	−	−	−	−	−	+	+	−	+	+	−	+	+	−	−	−
57989	16 June 2021	Blood	NI	XDR	277 *	+	−	−	−	−	−	+	+	+	+	+	−	+	+	−	−	−
58005	16 June 2021	CT	PI/PA	MDR		+	−	−	−	−	−	+	+	+	+	+	−	+	+	−	−	−
58007	16 June 2021	WS	NI	XDR	277 *	+	−	−	−	−	−	+	+	+	+	+	−	+	+	+	+	+
58111	30 June 2021	TS	NI	XDR	277 *	+	−	−	−	−	−	+	+	+	+	+	−	+	+	−	−	−
58218	15 July 2021	Urine	PI/PA	MDR		+	−	−	−	−	−	+	+	+	+	+	−	+	+	−	−	−
58276	27 July 2021	Liquor	PI/PA	MDR		+	−	−	−	−	−	+	+	+	+	+	−	+	+	−	−	−
58479	20 August 2021	Urine	NI	XDR	277 *	+	−	−	−	−	−	+	+	+	+	+	−	+	+	+	+	+
58482	20 August 2021	Blood	PR/PA	MDR		+	−	−	−	−	−	+	+	+	+	+	−	+	+	−	−	−
58608	10 September 2021	Urine	PI/PA	MDR		+	−	−	−	−	−	+	+	+	+	+	−	+	+	+	+	+
58739	28 September 2021	MT	PI/PA	MDR		+	−	−	−	−	−	+	+	+	+	+	−	+	+	+	+	+
58798	14 October 2021	TS	PR/PA	MDR		+	−	−	−	−	−	+	+	+	+	+	−	+	+	−	−	−
58820	14 October 2021	WS	PR/PA	MDR		+	−	−	−	−	−	+	+	+	+	+	−	+	+	−	−	−
58835	14 October 2021	IS	PR/PA	XDR	277 *	+	−	−	−	−	−	+	+	+	+	+	−	+	+	+	+	+
58924	28 October 2021	Urine	PI/PA	MDR		+	−	−	−	−	−	+	+	+	+	+	−	+	+	+	+	+
59035	18 November 2021	Blood	PI/PA	XDR	277 *	+	−	−	−	−	−	+	+	+	+	+	−	+	+	+	+	+
59183	10 December 2021	Urine	PI/PA	MDR		+	−	−	−	−	−	+	+	+	+	+	−	+	+	−	−	−
59202	10 December 2021	TS	PI/PA	XDR	277 *	+	−	−	−	−	−	+	+	+	+	+	−	+	+	−	−	−
59233	16 December 2021	TS	PI/PA	MDR		+	−	−	−	−	−	+	+	+	+	+	−	+	+	−	−	−
59329	5 January 2022	Urine	PR/PA	XDR	277 *	+	−	−	−	−	−	+	+	+	+	+	−	+	+	+	+	+

MDR: multidrug resistant; XDR: extensively drug resistant; PI: public institution, PR: private institution, PA: Pará State; AC: Acre State; NI: not informed; TS: tracheal secretion, LAS: lumbar abscess secretion; WS: wound secretion, IS: inguinal swab, MT: muscular tissue, CT: catheter tip; * defined by WGS; ^§^ defined by Sanger sequencing; +: positive; −: negative.

## Data Availability

All relevant data is provided within the manuscript.

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
