# Peer review of "Endemic High-Risk Clone ST277 Is Related to the Spread of SPM-1-Producing Pseudomonas aeruginosa during the COVID-19 Pandemic Period in Northern Brazil"

_microorganisms, 2023, doi:10.3390/microorganisms11082069_

Round 1

Reviewer 1 Report

This is a cross-sectional descriptive study whose aim was to provide information regarding antimicrobial resistance, virulence and genotypic features of SPM-1-producing P. aeruginosa isolates in the Northern region of Brazil. Although the paper overall is well written some information needs to be added to the manuscript:

1.       The name of the pathogens should be in italics. Please change.

2.       The Authors classify the isolates in MDR/XDR categories according to Magiorakos criteria which they cite in the article. However, recently, a new definition for resistant Pseudomonas aeruginosa such as Difficult to treat resistance Pseudomonas aeruginosa is being commonly used.  Could the Authors discuss in the introduction and discussion part also DTR- P. aeruginosa? Also if isolates showed characteristics of DTR-P. aeruginosa should be mentioned in the results.

3.       Limitation of the study in the discussion is missing. Please add.

4.       Could the Authors give information on the ward of isolation for each isolate?

5.       Resistance information to classic antipseudomonas antibiotics such as meropenem, and colistin and new antibiotics such as cefiderocol, ceftazidime-avibactam, ceftolozane-tazobactam is missing. Were they not tested? If not please add this information in the study limitations.

Author Response

Comments and Suggestions for Authors (Reviewer 1)

This is a cross-sectional descriptive study whose aim was to provide information regarding antimicrobial resistance, virulence and genotypic features of SPM-1-producing P. aeruginosa isolates in the Northern region of Brazil. Although the paper overall is well written some information needs to be added to the manuscript:

  1. The name of the pathogens should be in italics. Please change.

Reply: Corrections were performed as requested.

  1. The Authors classify the isolates in MDR/XDR categories according to Magiorakos criteria which they cite in the article. However, recently, a new definition for resistant Pseudomonas aeruginosasuch as Difficult to treat resistance Pseudomonas aeruginosa is being commonly used.  Could the Authors discuss in the introduction and discussion part also DTR- P. aeruginosa? Also, if isolates showed characteristics of DTR-P. aeruginosa should be mentioned in the results.

Reply: Difficult to treat resistance (DTR) Pseudomonas aeruginosa was classification as cited in the introduction, methods and discussion as requested. Strains presenting DTR classification were included in results (table 1).

  1. Limitation of the study in the discussion is missing. Please add.

Reply: Limitations of the study were included in the final part of discussion as requested.

  1. Could the Authors give information on the ward of isolation for each isolate?

Reply: Unfortunately, such information was not available in most of the forms sent to the laboratory, this is also reflected in the absence of information on the institutions of origin of some isolates as seen in table 2.

  1. Resistance information to classic antipseudomonas antibiotics such as meropenem, and colistin and new antibiotics such as cefiderocol, ceftazidime-avibactam, ceftolozane-tazobactam is missing. Were they not tested? If not please add this information in the study limitations.

Reply: The antimicrobials meropenem, colistin, and new antibiotics such as cefiderocol, ceftazidime-avibactam, ceftolozane-tazobactam were not tested solely due to unavailability of such antimicrobials in our laboratory. Moreover, it is our knowledge that antimicrobials such as cefiderocol, ceftazidime-avibactam, ceftolozane-tazobactam are still rarely applied in healthcare institutions in our region. Future work of our group vision to perform testing for such antimicrobials. Finally, the limitation pointed out was included as requested.

Reviewer 2 Report

This manuscript describes investigations of SPM producing P. aeruginosa ST277. Topic of this manuscript is an important issue, however, some modifications are necessary in the text.

1) In the title „Outbreak of SPM-1 producing Pseudomonas aeruginosa ST277…” is written, however, in the materials and methods part in line 96 this is written: „…antimicrobial resistance routine surveillance, from 2018 to 2021”.  

In case you report an outbreak in correlation with COVID-19, the time of isolation and duration of outbreak should be precisely given.Please, revise or clarify this!

2) Line 100: „In the present study, 34 non-repeatable isolates of P. aeruginosa obtained from various biological sample …” What was the time of isolation at these 34 isolates!

3) Some typing mistakes:

„illumina” should be written with capital: Illumina. Please check it and modify it in all over the manuscript.

„invitrogen”: should be written with capital: Invitrogen. Please check it and modify it in all over the manuscript.

4) Results: Line 229-230:  This list should be revised, because there are unnecessary  duplications: „aminoglycosides, Beta-Lactam, Carbapenem, Cephalosporin, Chloramphenicol, Fluoroquinolones, Fosfomycin, Gentamicin, Kanamycin. Streptomycin and Sulfonamide”

I suggest this form: Carbapenem, Cephalosporin, Chloramphenicol, Fluoroquinolones, Fosfomycin, Gentamicin, Kanamycin, Streptomycin and Sulfonamide

5) Figure 1 should be enlarged. it is difficult to see in its current form.

6) Table 2 is also difficult to see. Please, revise it to get to a better view.

Quality of English is good. Only some minor modifications are necessary.

Author Response

Comments and Suggestions for Authors (Reviewer 2)

 This manuscript describes investigations of SPM producing P. aeruginosa ST277. Topic of this manuscript is an important issue, however, some modifications are necessary in the text.

 1) In the title „Outbreak of SPM-1 producing Pseudomonas aeruginosa ST277…” is written, however, in the materials and methods part in line 96 this is written: „…antimicrobial resistance routine surveillance, from 2018 to 2021”.  In case you report an outbreak in correlation with COVID-19, the time of isolation and duration of outbreak should be precisely given.Please, revise or clarify this!

Reply:  We used the word 'outbreak' considering the high similarity of the isolates as verified by the WGS methodology and because they share ST277. However, the authors of the present work chose to revise the use of the term due to the absence of some critical information such as outbreak duration. Thus, the text was modified as requested. The changes were also extended to the title of the paper: "Endemic high-risk clone ST277 is related to the spread of SPM-1-producing Pseudomonas aeruginosa during the COVID-19 pandemic period in the Brazilian Northern region"

2) Line 100: „In the present study, 34 non-repeatable isolates of P. aeruginosa obtained from various biological sample …” What was the time of isolation at these 34 isolates!

Reply: P. aeruginosa isolates were recovered from 2018 to 2021. Text was modified as requested.

3) Some typing mistakes:

„illumina” should be written with capital: Illumina. Please check it and modify it in all over the manuscript.

„invitrogen”: should be written with capital: Invitrogen. Please check it and modify it in all over the manuscript.

Reply: The text was modified as requested.

4) Results: Line 229-230:  This list should be revised, because there are unnecessary  duplications: „aminoglycosides, Beta-Lactam, Carbapenem, Cephalosporin, Chloramphenicol, Fluoroquinolones, Fosfomycin, Gentamicin, Kanamycin. Streptomycin and Sulfonamide”

I suggest this form: Carbapenem, Cephalosporin, Chloramphenicol, Fluoroquinolones, Fosfomycin, Gentamicin, Kanamycin, Streptomycin and Sulfonamide

Reply: The text was modified as requested.

5) Figure 1 should be enlarged. it is difficult to see in its current form.

Reply: Figure 1 as modified and corrected as requested by authors and reviewer.

6) Table 2 is also difficult to see. Please, revise it to get to a better view.

Reply: Table 2 as modified as requested by authors and reviewer. We would also like to point out that the best arrangement of the table will be discussed together with the editorial team of the journal.